# Facile Fabrication of a Novel PZT@PPy Aerogel/Epoxy Resin Composite with Improved Damping Property

**DOI:** 10.3390/polym11060977

**Published:** 2019-06-03

**Authors:** Chunmei Zhang, Yuchao Li, Yanhu Zhan, Qian Xie

**Affiliations:** School of Materials Science and Engineering, Liaocheng University, Liaocheng 252059, China; zhangchunmei@lcu.edu.cn (C.Z.); zhanyanhu@lcu.edu.cn (Y.Z.); xieqian@lcu.edu.cn (Q.X.)

**Keywords:** aerogel, epoxy, damping property, composite

## Abstract

A novel lead zirconate titanate@polypyrrole (PZT@PPy) aerogel (PPA) was fabricated via in-situ polymerization and subsequent freeze-drying method. The porous PPA was then saturated with epoxy resin to obtain the PPA/epoxy composite (PPAE) by a simple vacuum filling method. In this way, the filler content and dispersion uniformity are well guaranteed, which is in favor of improving the damping and mechanical properties of composites. The morphology and structure of PPAs were investigated using XRD, SEM, EDS and nitrogen absorption and desorption measurements. The results showed that the PPA possessed a three-dimensional porous structure with uniform lead zirconate titanate (PZT) distribution. The influence of PZT content on the damping property of PPAE composite was investigated by dynamic mechanical analysis (DMA). PPAE-75 (i.e., the mass ratio of PZT to PPy is 75 wt %) exhibited the maximum damping loss factor value, 360% higher than that of the epoxy matrix, suggesting good structural damping performance.

## 1. Introduction

With the rapid development of modern industry and traffic system, vibration and noise pollution, which is harmful to industrial safety, human health and ecological environment, has become increasingly serious. The techniques used to reduce vibration and noise have attracted much more attention in modern engineering fields [1,2,3]. Nowadays, polymers have shown great potential for damping due to their excellent viscoelasticity and good processability. Epoxy resin is one of the most widely used polymers in fields of coatings, adhesives and structural materials, moreover, fiber reinforced epoxy composites are usually applied as high-performance structural materials in aviation and aerospace industry due to their low density and good mechanical performance [4,5]. Since working environments with epoxy as a structural material are mostly under room conditions, it is important to improve the damping property of epoxy at room temperature. 

Considerable efforts have been made to improve the room temperature damping capability of epoxy resin by adding various nanofillers. Rajoria et al. [6] fabricated carbon nanotube-epoxy composites, and an increase of up to 700% in damping ratio was observed by using 5.0 wt % multi-walled carbon nanotubes (MWCNTs) in epoxy. Khan et al. [7] studied the vibration damping properties of carbon fiber-reinforced epoxy containing multiwall carbon nanotubes in different amplitudes, natural frequencies and vibration modes. The results showed that the damping ratios of the composites increased with increasing carbon nanotube (CNT) content. The damping performance of the polymer composites is enhanced with the addition of CNTs, which is attributed to the “stick-slip” interfacial friction mechanism as described by Zhou et al. [8]. The room temperature damping performance of epoxy can be improved by adding nanofillers through the sliding at filler–matrix interfaces.

The piezo-damping effect is a novel method to improve the room temperature damping property of materials. Piezo-damping material is composed of a piezoelectric phase, a conductive phase and the polymer matrix. Since it was proposed by Sumita in 1991 [9], various piezo-damping composites have been studied and some promising results have already been obtained. Hori et al. [10] fabricated a composite composed of PZT powders, carbon black and epoxy resin, and it was found that a maximum damping loss factor of 0.08 could be obtained at room temperature compared with 0.035 of the EP matrix. Tanimoto et al. [11] synthesized PZT/carbon fiber/epoxy composites, showing a damping ratio of about 0.02 at room temperature. Skandani et al. [12] prepared ZnO nanorod/carbon fiber/epoxy composites, exhibiting a room temperature loss factor of about 0.015. Carponcin et al. [13] fabricated PZT/carbon nanotubes/polyamide 11 composites, obtaining a loss factor of about 0.018 at room temperature. These reports proved that the damping behavior of polymers can be improved by the introduction of piezo-damping effect. Briefly speaking, the external mechanical energy, like vibration and noise, could be transformed into electrical energy through the piezoelectric effect of the piezoelectric ceramics, and then the generated electrical energy can be dissipated into heat as it flows through the resistive phase of the composite [14,15,16]. To guarantee the generated electrical energy fully dissipated, the volume resistivity of the material should be better adjusted in the semiconductor range.

Previous research mainly involved adding various fillers in the polymer matrix by simple mechanical mixing. The content and dispersion uniformity of the fillers can be limited to a certain extent, especially when higher filler loading is required, which can have an adverse effect on the damping and mechanical properties of composites. In this paper, porous PZT@PPy aerogel was fabricated via in-situ polymerization and then saturated with epoxy to obtain the PPA/epoxy composite. In this way, the content of the filler is unlimited to a large extent, which can produce abundant interfacial friction. Moreover, the dispersion uniformity is also well guaranteed, which is in favor of the improvement of the composites’ damping and mechanical properties. In this paper, the morphology and structure of PPAs are studied, and the damping property of PPAE composite is investigated and discussed. 

## 2. Experimental Section

### 2.1. Materials

The pyrrole monomer (Py, AR, 99%), ferric chloride hexahydrate (FeCl_3_·6H_2_O, AR, 99%), acetone (AR, 99.5%), curing agent 4,4′-diaminodiphenylmethane (DDM), and ethyl alcohol (AR, 99.7%) were purchased from Sinapharm Chemical Reagent Co., Ltd., Shanghai, China and all the chemicals were used without further purification. The epoxy resin used was E51 and it was bought from Shanghai Resin Factory Co., Ltd., Shanghai, China. The PZT powders with several microns in diameter were purchased from ZiBo Bailing Functional Ceramics Co., Ltd., ZiBo, China and the value of its piezoelectric coefficient (d_33_) before balling into powders was approximately 650 pC/N. 

### 2.2. Preparation of PZT@PPy Aerogels (PPAs)

PZT@PPy aerogels with different PZT loadings were prepared as shown schematically in Figure 1. For the fabrication of PPA-25 (i.e., the mass ratio of PZT to PPy is 25 wt %), a certain amount of PZT powders (0.17 g) were added to a solution of PPy monomer (0.67 g) and ethanol (6 mL), followed by stirring in ice bath for 30 min to get a mixture A. Meanwhile, 6.2 g FeCl_3_·6H_2_O was dissolved in 6 mL deionized water and stirred for several minutes to get a solution B. In an ice bath, solution B was added into mixture A dropwise, under rapid stirring for approximately 10 min and a black hydrogel was formed. The gel was then aged at room temperature for 24 h to further complete the polymerization reaction of pyrrole monomers [17]. Afterwards, the gel was washed with deionized water and alcohol several times to remove residual ferric chloride and freeze-dried for 48 h to get PPA-25 aerogel. Similarly, changing the PZT content to 0, 0.34, 0.50 and 0.67 g (the mass ratio of PZT to PPy is 0, 50, 75 and 100 wt % respectively), PPA-0, PPA-50, PPA-75 and PPA-100 aerogels can be obtained. 

### 2.3. Fabrication of PZT@PPy Aerogel/Epoxy Composites (PPAEs)

The epoxy E51 was preheated to 70 °C for about 30 min, so its viscosity was lowered and can be weighed precisely. A certain amount of curing agent DDM was dissolved in acetone and heated at 70 °C to obtain a clear solution. Then a certain amount of E51 was diluted with acetone and mixed uniformly with the DDM solution to get the E51 mixture. Subsequently, the E51 mixture was added dropwise into the PPAs and then placed under vacuum to remove the acetone solvent and bubbles trapped in the composite. This process was repeated several times until the aerogel was fully saturated with epoxy. Afterwards, the composite was cured at 80 °C for 2 h, 120 °C for 2 h and 160 °C for 2 h to finally get PPAEs. Composite PPAEs with different amounts of PZT ceramics were named as PPAE-0, PPAE-25, PPAE-50, PPAE-75 and PPAE-100 respectively. The mass ratio of EP to DDM in the experiment was held constant at 4:1.

### 2.4. Characterization and Testing

X-ray diffraction (XRD) spectra were acquired by D/MAX2550/PC (Rigaku Corporation, Tokyo, Japan) using Cu Kα radiation from 20° to 70° at a scan rate of 5°·min^−1^ under 35 kV and 200 mA. The microstructure and the elemental analysis of PPAs were characterized by field-emission scanning electron microscopy (FE-SEM, Hitachi S-4800, Hitachi Limited, Tokyo, Japan) and energy dispersive spectroscopy (EDS, Hitachi S-4800, Hitachi Limited, Tokyo, Japan). X-ray photoelectron spectroscopy (XPS) was recorded using a Kratos Axis Ultra DLD spectrometer (Shimadzu Corporation, Tokyo, Japan). Nitrogen absorption and desorption measurements were performed with an Auto sorb IQ instrument (Beishide Instrument Technology Co., Ltd, Beijing, China). The surface areas were calculated by the Brunauer-Emmett-Teller (BET) method. Volume resistivity was measured using a ZC-36 high resistance meter from Sixth Electric Meter Factory of Shanghai, Shanghai, China. Dynamic mechanical measurements were performed on Perkin-Elmer DMA 8000 (Perkin-Elmer, MA, USA) and rectangular specimens of 30 × 8 × 2 mm were used for testing. The material property measurements were carried out under a three-point bending mode at the frequency of 1 Hz. The temperature range was from 0 to 160 °C at a heating rate of 5 °C/min and the storage modulus, loss modulus, and loss factor were obtained simultaneously. 

## 3. Results and Discussion

Figure 2a,b are the SEM images of the PPy aerogel. It can be found that the aerogel shows a three-dimensional porous structure with a pore diameter in the micron range. The skeleton of the aerogel is composed of polymerized polypyrrole nanoparticles, which are self-assembled together through the π-π complexation between PPy rings. The SEM pictures of the PZT@PPy aerogels with different amount of PZT are shown in Figure 2c,d, and Appendix A. It can be found that the PPAs retained the three-dimensional porous structure, and the frameworks are constituted by uniformly distributed PZT ceramics and PPy nanoparticles. Moreover, with the PZT content increased, piezoelectric ceramics gradually become the major component of the PPAs (Figure 2c, Appendix A). Furthermore, the surface of the PZT ceramics are almost covered by continuously connected PPy nanoparticles (Appendix A), which is beneficial for the external energy dissipation via the piezo-damping effect. The photos of the prepared PZT@PPy hydrogel and aerogel are given in Figure 2f. Depending on the vessels used, the aerogels can be fabricated into any shape to meet specific requirements in practical applications. 

Figure 3 shows the EDS elemental mapping (C, N, O, Zr, Ti) of PPA-50. The C and N elements are attributed to PPy nanoparticles, and the elements O, Zr and Ti are from PZT ceramics. The results prove the coexistence of the two components and further confirm the uniform distribution of PZT ceramics in the aerogel. The PPAE was then fabricated by saturating the PPA with epoxy polymer. As shown in Figure 2e, the PPAs maintain their three-dimensional network structure in the filling process, ensuring that the piezoelectric PZT ceramics and conductive PPy nanoparticles can be evenly dispersed in the epoxy matrix. 

The nitrogen adsorption-desorption isotherm and the matching DFT (Density functional theory) pore size distribution plot of the PPAs with different PZT content are shown in Figure 4 and Appendix A, and the results are listed in Appendix A. It is found that the nitrogen adsorption isotherms for the PPA-75 are Type II, indicating an interconnected mesoporous system with a broad pore-size distribution [18,19]. The increased PZT contents do not have a remarkable influence on the porous structure of the PPAs, however, resulting in a gradual decrease of the BET surface area from 32.100 to 12.663 m^2^/g and the cumulative pore volume from 0.102 to 0.045 cm^3^/g (Appendix A) for PPA-0 and PPA-100 respectively. This can be attributed to that as the PZT content increased, more pores and channels of the aerogels were blocked. The DFT analysis shows that the PPAs possess average mesopores of approximately 4 nm, which are formed through the accumulation of PPy nanoparticles.

Figure 5 is the XPS spectra for PPA-75. The peaks, with a binding energy of 284.77 and 400.02 eV, originate from the C 1s and N 1s, respectively. The C 1s peak can be reasonably decomposed into two Gaussian peaks with a binding energy of 288.93 and 284.83 eV, which can be attributed to C–N and C=C, respectively (Figure 5b). The N 1s peak can be divided into three Gaussian peaks with binding energy of 398.48, 399.98 and 401.63 eV, respectively (Figure 5c), which are from the neutral and imine-like structure (−C=N−), the neutral and amine-like structure (−N−H−) and positive charged nitrogen atoms (−NH^+^−) of the PPy nanoparticles as described in other reports [20,21]. Moreover, the peak corresponding to the binding energy of 198.08 eV belongs to the element Cl, which is generated from the residual oxidant ferric chloride. In addition, the peaks for O 1s, Zr 3d, Ti 2p and Pb 4f are also observed in Figure 5a, which confirm the successful fabrication of PPA.

The XRD patterns of the PPy aerogel and PPAs are shown in Figure 5d. The XRD result of PPy aerogel shows a broad band near 22.16° on account of the amorphous structure of PPy polymer [22]. For PPAs, the XRD spectra exhibit both the PPy and the main PZT peaks, which confirm the coexistence of the PPy conducting polymer and the PZT ceramics in the composite aerogel. These results also suggest that the PZT ceramics retain their perovskite structure in the fabrication process.

The volume resistivity of PPAEs with different PZT content is shown in Figure 6. It can be found that the volume resistivity for PPAE-0, PPAE-25, PPAE-50, PPAE-75 and PPAE-100 are 130.9, 61.5, 31.1, 19.2 and 10.9 MΩ respectively, which are all adjusted to the semiconductor range and benefit the effective function of piezo-damping effect. 

Figure 7a shows the storage modulus (E’) values of the epoxy and PPAEs, and high E’ value means high stiffness of a material. As shown in Table 1, the E’ values of PPAEs at room temperature are improved to some extent than that of the epoxy resin, except for the PPAE-100, which shows slightly lower E’ value than the polymer matrix. For PPAEs with PZT content lower than 100 wt %, higher E’ values may be attributed to as following: on one hand, the functional groups on the surface of PZT or PPy particles, like hydroxyl or amino groups, may react with the epoxy, thus resulting in strong interfacial bonding between the fillers and the polymer matrix—on the other hand, the addition of the high modulus PZT ceramics is also favorable to improve the stiffness of the PPAEs [23,24]. For composite of PPAE-100, with a PZT content increased to 100 wt %, the pore volume and pore size of PPA-100 exhibit an obvious decrease as listed in Appendix A, which leads to fewer filling of the polymer matrix, thus resulting in lower stiffness. 

Figure 7b shows the loss modulus (E’’) values of the epoxy and PPAEs. As shown in Table 1, the E’’ values for the epoxy, PPAE-0, PPAE-25, PPAE-50, PPAE-75 and PPAE-100 at 20 °C are 79.8, 177.6, 261.8, 324.5, 413.9 and 137.2 MPa, respectively. The E’’ values of PPAEs are improved correspondingly with increasing PZT content, reaching a maximum value for PPAE-75, which shows an increase of about 419% compared with the epoxy matrix. As the PZT loading continues increasing to 100 wt %, the E’’ shows a significant decrease but is still higher than the epoxy. The results exhibit that all the PPAEs possess higher E’’ values than the epoxy matrix, indicating that the PPAEs can dissipate more mechanical vibration and noise into heat energy. 

The loss factor (tanδ) values of the epoxy and PPAEs are shown in Figure 7c. Normally, the service environment of most engineering materials is near room temperature, hence it is significant to improve the tanδ value near room condition. As shown in Table 1, the PPAEs all exhibit higher tanδ values than the epoxy, and it gradually increases with increasing PZT content. PPAE-75 shows the best damping performance, with an increase of approximately 360% compared to the polymer matrix. As the PZT content increases to 100 wt %, tanδ value displays an obvious decrease for PPAE-100. 

The improvement of the damping performance of PPAEs is attributed to the piezo-damping effect and the internal friction effect. When PPAE composite is subjected to an external alternating force, some strain energy can be transformed into electrical energy via the piezoelectric effect of the PZT ceramics, and then the generated electrical current is dissipated into heat when flowing through the PPy conductive networks. Furthermore, there is abundant interfacial contact between the fillers (PPy nanoparticles and PZT ceramics) themselves and between the fillers and the matrix, thus some mechanical energy can be dissipated into heat via frictions from boundary sliding (filler–filler) and interfacial sliding (filler–matrix) [8,9,10]. The internal friction energy dissipation effect can also be proved by the increased damping performance of PPAE-0, which containing no PZT piezoelectric ceramics. For composite PPAE-100, since the energy dissipation through frictions between fillers and matrix is greatly reduced due to the limited polymer filling, the damping capacity shows a decreased trend.

## 4. Conclusions 

A novel PZT@PPy aerogel was prepared by in-situ polymerization and freeze-drying method, and then it was saturated with epoxy to obtain PPAE composite via a facile vacuum filling method. In this way, the filler content and dispersion uniformity are well guaranteed, which is beneficial for the improvement of composites’ damping and mechanical properties. PPAE-75 showed the best damping performance, with a tanδ value increasing by about 360% compared with the epoxy matrix, which is due to the piezo-damping and internal friction effect. The results indicate that the PPAEs can be used as good structural damping materials.

## Figures and Tables

**Figure 1 polymers-11-00977-f001:**
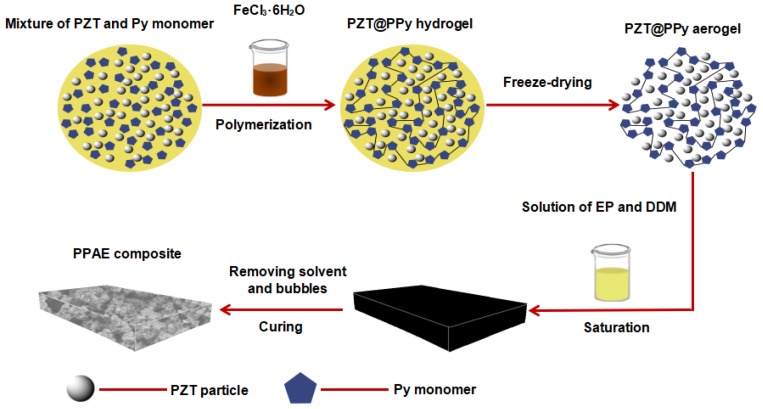
The fabrication process of lead zirconate titanate@polypyrrole (PZT@PPy) aerogel (PPA)/epoxy composite (PPAE).

**Figure 2 polymers-11-00977-f002:**
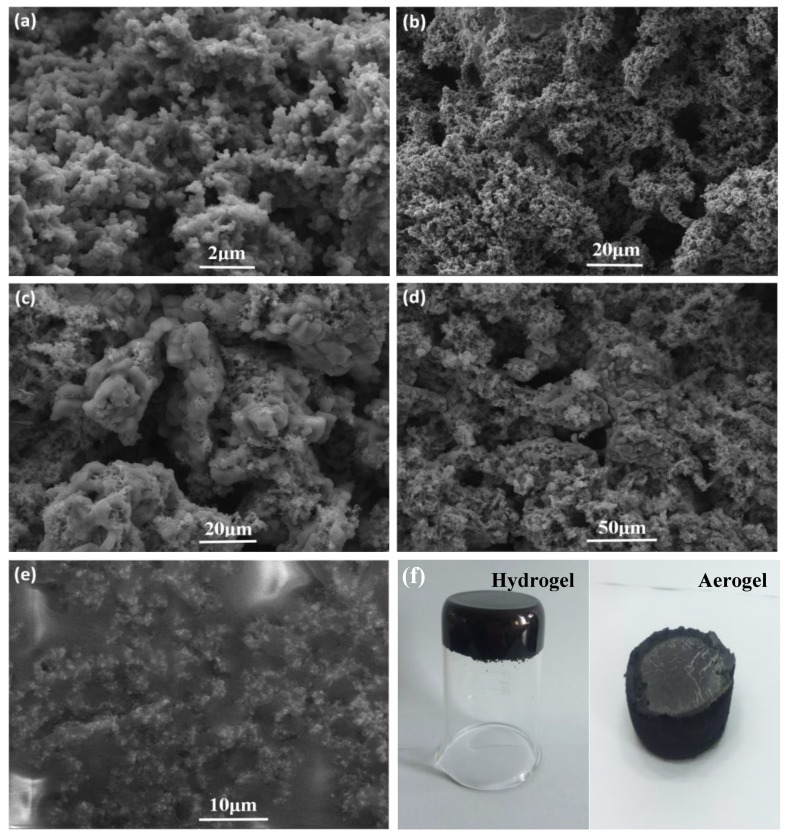
The SEM images of (**a**) and (**b**) PPy aerogel, (**c**) and (**d**) PPA-75, and (**e**) PPAE-75; (**f**) the photos of the prepared PZT@PPy hydrogel and aerogel.

**Figure 3 polymers-11-00977-f003:**
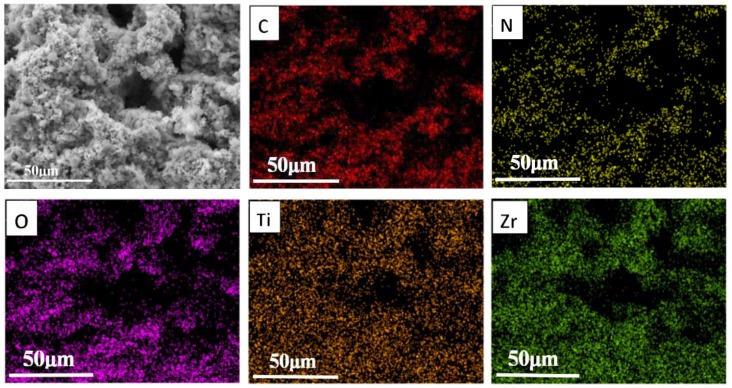
The element mapping images of the PPA-50.

**Figure 4 polymers-11-00977-f004:**
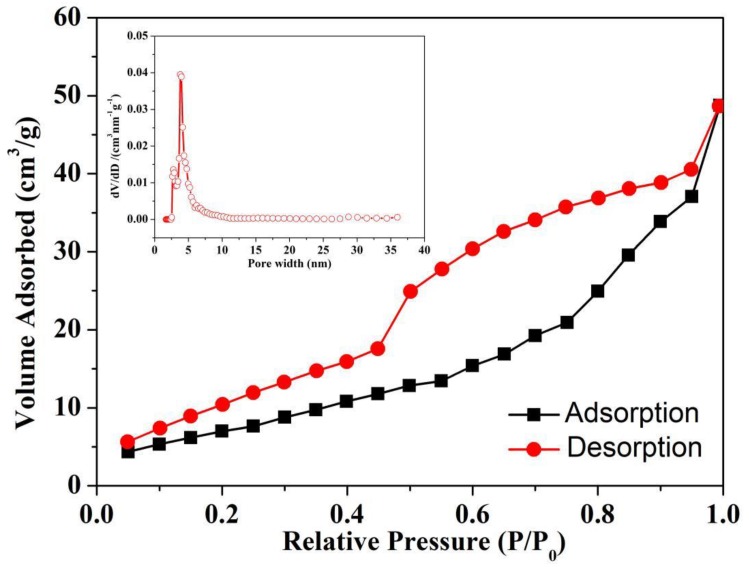
The nitrogen adsorption and desorption isotherm and the pore size distribution plot of PPA-75.

**Figure 5 polymers-11-00977-f005:**
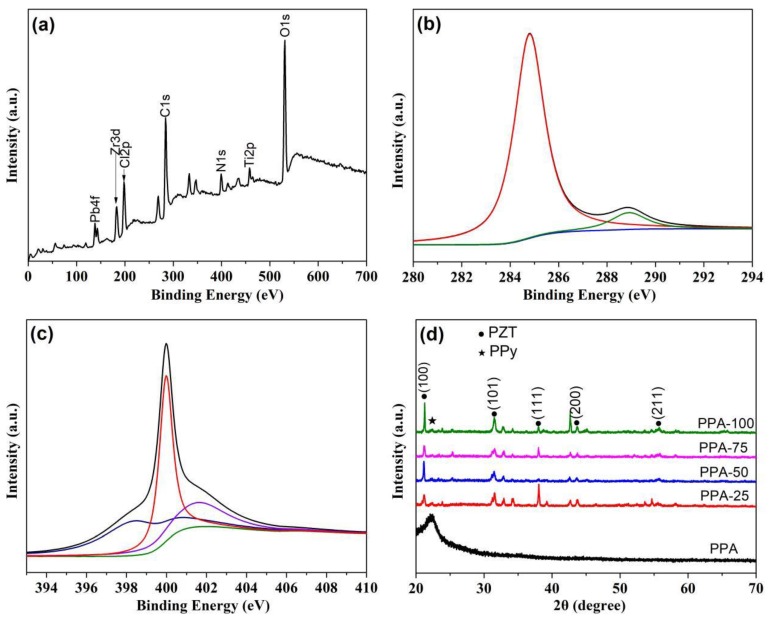
(**a**) Full XPS spectrum, (**b**) C 1s and (**c**) N 1s XPS core-level spectra of PPA-75; (**d**) XRD spectra of PPy and PPAs with different content of PZT ceramics.

**Figure 6 polymers-11-00977-f006:**
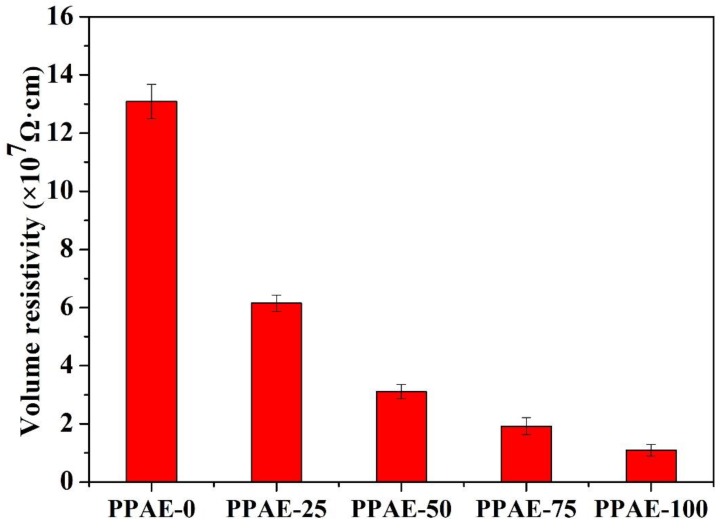
The volume resistivity values of PPAEs with different content of PZT ceramics.

**Figure 7 polymers-11-00977-f007:**
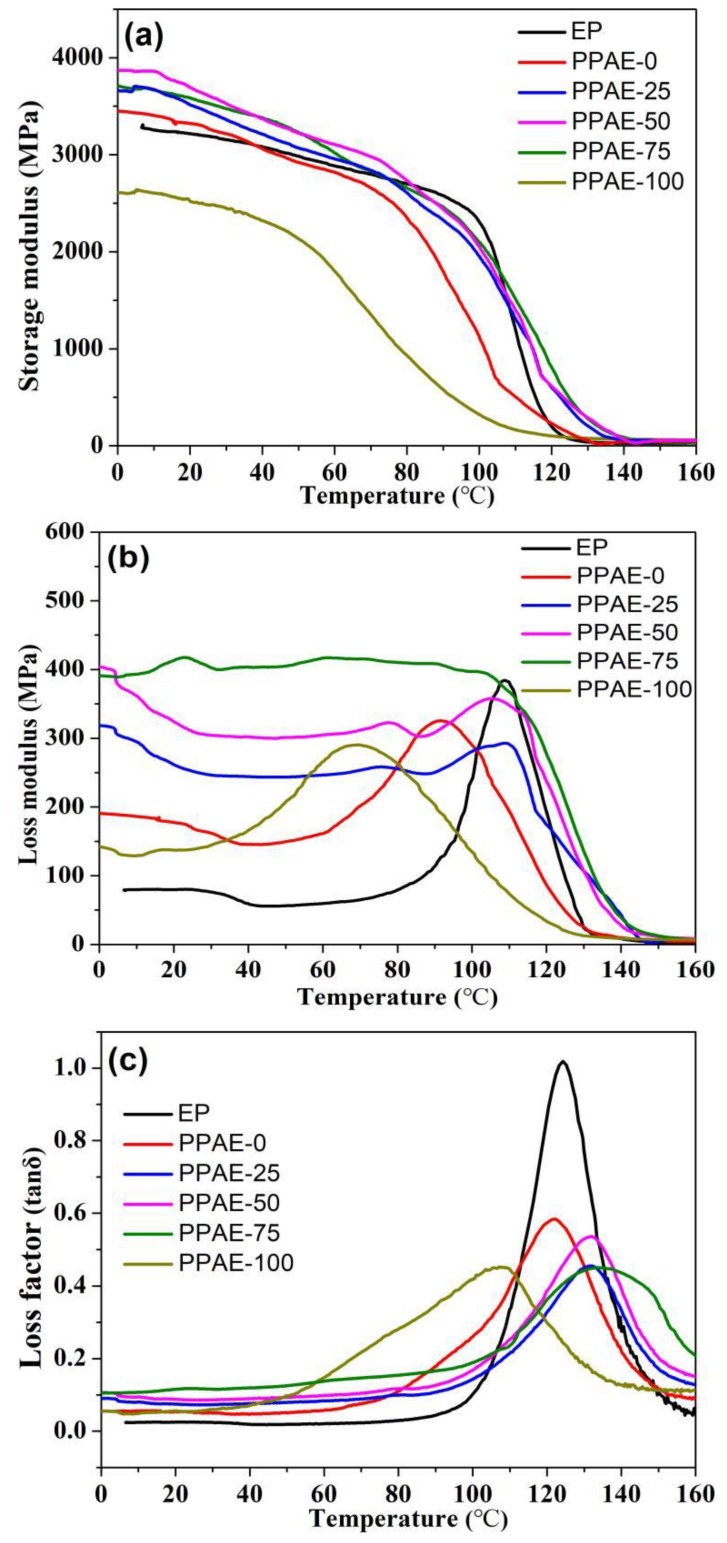
The variation plots of (**a**) the storage modulus, (**b**) loss modulus and (**c**) loss factor as a function of temperature (0–160 °C) for epoxy matrix and the PPAEs.

**Table 1 polymers-11-00977-t001:** Influence of different PZT content on damping behavior of PPAEs from 0–160 °C.

Sample (at 20 °C)	Storage Modulus (MPa)	Loss Modulus (MPa)	Loss Factor	Tg (°C)	Temperature Range (∆T/tanδ > 0.3) (°C)
Epoxy	3215.2	79.8	0.025	124.3	109.9–139.7 (29.8)
PPAE-0	3322.5	177.6	0.053	122.2	103.9–136.1 (32.2)
PPAE-25	3516.6	261.8	0.074	132.0	118.9–141.3 (22.4)
PPAE-50	3696.0	324.5	0.088	132.2	114.3–143.1 (28.8)
PPAE-75	3584.9	413.9	0.115	133.9	115.8–152.4 (36.6)
PPAE-100	2522.3	137.2	0.040	107.1	83.7–119.3 (35.6)

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
