# Peer review of "Facile Fabrication of a Novel PZT@PPy Aerogel/Epoxy Resin Composite with Improved Damping Property"

_polymers, 2019, doi:10.3390/polym11060977_

Round 1
Reviewer 1 Report
This paper introduces the facile vacuum filling method for fabricating the PZT@PPY aerogel/epoxy resin composite, followed by characterization and testing such as X-ray diffraction spectra, microstructure and elemental analysis, X-ray photoelectron spectroscopy, nitrogen adsorption and desorption measurements, surface areas, volume resistivity, and dynamic mechanical measurements. The PPAE-75 was found to have the best damping performance and could be used as good structural damping materials. A few comments for the authors before this paper can be accepted for publication.
1. Page 1, Lines 34 and 37: “MWCNT” and “CNT” please list the full names of terminologies before using abbreviations.
2. Page 2, Lines 46-47: “… and it was found that at room temperature a maximum damping loss factor of 0.08 could be obtained at room temperature …” “at room temperature” repeated.
3. Please avoid using the first person (e.g., “we”) in a technical paper.
4. Page 4, Line 123: There is no Figure S1 in this paper, same below.
5. Page 4, Fig. 2 (f): Please label which one is the hydrogel and which one is the aerogel.
6. Page 5, Line 144: What is “DFT”?
7. Page 5, Line 145: There is no Figure S2 in the paper.
8. Page 5, Line 146: There is no Table S1 in the paper, same below.
Author Response
Dear Reviewer,
Thank you for your comments concerning our manuscript entitled “Facile fabrication of a novel PZT@PPy aerogel/epoxy resin composite with improved damping property” (ID: polymers-511226). We have studied comments carefully and have made correction which we hope meet with approval. The response to the comments are as following:
Reviewer 1:
1. Response to comments: Page 1, Lines 34 and 37: “MWCNT” and “CNT” please list the full names of terminologies before using abbreviations.
Response: Thanks for reviewer’s suggestions. The full names of Multi-walled carbon nanotubes and carbon nanotube have been added respectively before the abbreviations of “MWCNTs” and “CNT”.
2. Response to comments: Page 2, Lines 46-47: “… and it was found that at room temperature a maximum damping loss factor of 0.08 could be obtained at room temperature …” “at room temperature” repeated.
Response: Thanks for reviewer’s suggestions. We have revised this problem, and the correct description is that it was found that a maximum damping loss factor of 0.08 could be obtained at room temperature compared with 0.035 of the EP matrix.
3. Response to comments: Please avoid using the first person (e.g., “we”) in a technical paper.
Response: Thanks for reviewer’s suggestions. We have corrected this statement, and there has been no first person term in the manuscript.
4. Response to comments: Page 4, Line 123: There is no Figure S1 in this paper, same below.
Response: Thanks for reviewer’s suggestions. We have added Figure S1 in the Supporting information file.
5. Response to comments: Page 4, Fig. 2 (f): Please label which one is the hydrogel and which one is the aerogel.
Response: Thanks for reviewer’s suggestions. The hydrogel and aerogel have been labeled in Figure 2(f).
6. Response to comments: Page 5, Line 144: What is “DFT”?
Response: Thanks for reviewer’s questions. DFT means Density Functional Theory, and it is a method to measure the pore size distribution of porous materials. The explanation has been added in the manuscript.
7. Response to comments: Page 5, Line 145: There is no Figure S2 in the paper.
Response: Thanks for reviewer’s suggestions. We have added Figure S2 in the Supporting information file.
8.Response to comments: Page 5, Line 146: There is no Table S1 in the paper, same below.
Response: Thanks for reviewer’s suggestions. We have added Table S1 in the Supporting information file.
Special thanks to you for your good comments.

Reviewer 2 Report
A study on a new combination of PZT material and polymer for the damping composite is presented in a consistent way and well analysed.
The following corrections are suggested:
1) Figure 2 - Please made it at the same scale for comparison purposes. (change a), d), e), etc.)
2) Page 5 Line 147-148: On mesoporous systems and adsorption J.C.Groen,L.A.A. Peffer and J.Pérez-Ramirez, Microp. Mesop. Mater, 60 (2003) 1
3) Please, describe in details (could be added as a part of SI) how the actual samples were prepared starting from aerogel/composite for different measurements (BET, SEM and so on). Sectioning, homogenization etc.
4) Minor: correct degree sign through the article so it will be a degree sign.
Author Response
Dear Reviewer,
Thank you for your comments concerning our manuscript entitled “Facile fabrication of a novel PZT@PPy aerogel/epoxy resin composite with improved damping property” (ID: polymers-511226). We have studied comments carefully and have made correction which we hope meet with approval. The response to the comments are as following:
Reviewer 2:
1. Response to comments: Figure 2 - Please made it at the same scale for comparison purposes. (change a), d), e), etc.)
Response: Thanks for reviewer’s suggestions. Figure 2(a) and (b) are the SEM images of PPy aerogel, and Figure 2(c) and (d) are the SEM images of PPA-75. SEM pictures of different magnifications were used to more clearly show the microstructure and morphology of PPy aerogel and PPA-75. The skeleton of the PPy aerogel is composed of polymerized polypyrrole nanoparticles, and the PPA-75 was composed of PPy nanoparticles and PZT ceramics with diameter of several microns, so different scale bar were used to ensure a clear exhibition of the microstructure and morphology of the aerogels with or without PZT ceramics.
2. Response to comments: Page 5 Line 147-148: On mesoporous systems and adsorption J.C.Groen,L.A.A. Peffer and J.Pérez-Ramirez, Microp. Mesop. Mater, 60 (2003) 1
Response: Thanks for reviewer’s suggestions. The nitrogen adsorption-desorption isotherm and the pore size distribution were used to analyze the changes of pore volume and pore size distribution of the PPAs. The recommended article has been cited in the corresponding part of the manuscript.
3. Response to comments: Please, describe in details (could be added as a part of SI) how the actual samples were prepared starting from aerogel/composite for different measurements (BET, SEM and so on). Sectioning, homogenization etc.
Response: Thanks for reviewer’s suggestions. The PPAs were cut into pieces, and the aerogel blocks were used directly for BET and SEM measurement. After saturating the PPAs with epoxy resin, the PPAEs were obtained. The PPAE composite were cut into rectangular specimens of 10 × 10 × 2 mm for SEM measurement. The PPAE composite were cut into rectangular specimens of 60 × 60 × 2 mm for volume resistivity measurement using a ZC-36 high resistance meter. The PPAE composite were cut into rectangular specimens of 30 × 8 × 2 mm for Dynamic mechanical measurements by Perkin-Elmer DMA 8000. The description has been added in the Supporting information file.
4. Response to comments: Minor: correct degree sign through the article so it will be a degree sign
Response: Thanks for reviewer’s suggestions. The degree sign through the article has been all corrected according to your requirement.
Special thanks to you for your good comments.

Reviewer 3 Report
The paper describes an interesting work concerning the possibility of fabricating a novel PPA aerogel. All the aspects are critically discussed in the Experimental and Results sections. I think the paper could be accept for publication after some minor spell check in the grammar and some additional papers in the Introduction Section.
Author Response
Dear Reviewer,
Thank you for your comments concerning our manuscript entitled “Facile fabrication of a novel PZT@PPy aerogel/epoxy resin composite with improved damping property” (ID: polymers-511226). We have studied comments carefully and have made correction which we hope meet with approval. The response to the comments are as following:
Reviewer 3:
The paper describes an interesting work concerning the possibility of fabricating a novel PPA aerogel. All the aspects are critically discussed in the Experimental and Results sections. I think the paper could be accept for publication after some minor spell check in the grammar and some additional papers in the Introduction Section.
Response: Thanks for reviewer’s suggestions. The spelling and grammar problems of the manuscript have been carefully checked and the mistakes have been corrected, and some papers related to piezo-damping materials have been added in the Introduction Section.
Special thanks to you for your good comments.

Round 2
Reviewer 1 Report
I have no more comments on this paper and I think it can be accepted for publication.